# Persistent Current of SU(*N*) Fermions

Wayne J. Chetcuti,[1,2,3] Tobias Haug,[4] Leong-Chuan Kwek,[4,5] and Luigi Amico[3,4,6,∗]

[1]*Dipartimento di Fisica e Astronomia, Via S. Sofia 64, 95127 Catania, Italy*
[2]*INFN-Sezione di Catania, Via S. Sofia 64, 95127 Catania, Italy*
[3]*Quantum Research Centre, Technology Innovation Institute, Abu Dhabi, UAE*
[4]*Centre for Quantum Technologies, National University of Singapore, 3 Science Drive 2, Singapore 117543, Singapore*
[5]*MajuLab, CNRS-UNS-NUS-NTU International Joint Research Unit, UMI 3654, Singapore*
[6]*LANEF 'Chaire d'excellence', Universitè Grenoble-Alpes & CNRS, F-38000 Grenoble, France*
(Dated: January 17, 2022)

We study the persistent current in a system of SU(*N*) fermions with repulsive interaction, confined in a ring-shaped potential and pierced by an effective magnetic flux. Several surprising effects emerge. As a combined result of spin correlations, (effective) magnetic flux and interaction, spinons can be created in the ground state such that the elementary flux quantum can change its nature. The persistent current landscape is affected dramatically by these changes. In particular, it displays a universal behaviour. Despite its mesoscopic character, the persistent current is able to detect a quantum phase transition (from metallic to Mott phases). Most of, if not all, our results could be experimentally probed within the state-of-the-art quantum technology, with neutral matter-wave circuits providing a particularly relevant platform for our work.

Quantum technology intertwines basic research in quantum physics and technology to an unprecedented degree: different quantum systems, manipulated and controlled from the macroscopic spatial scale down to individual or atomic level, can be platforms for quantum devices and simulators with refined capabilities; on the other hand, the acquired technology prompts new studies of fundamental aspects of quantum science with an enhanced precision and sensitivity. Amongst the various quantum systems relevant for quantum technologies, ultracold atomic systems play an important role due to their excellent coherent properties and enhanced control and flexibility of the operating conditions [1]. Atomtronics is an emerging research area in quantum technology exploiting cold atoms matter-wave circuits with a variety of different architectures [2]. Being characterized by distinctive physical principles, atomic circuits can define a quantum technology with specific features. In particular, one of the peculiar knobs that can be exploited in atomtronics is the statistics of the particles forming the quantum fluid flowing in the circuit. Most of the studies carried out so far have been devoted to atomtronic circuits of ultracold bosons, whilst ones comprised of interacting ultracold fermions require extensive investigations.

In this paper, we focus on quantum fluids comprising of interacting multicomponent spin SU(*N*) fermions. Strongly interacting fermions with *N* spin components, as provided by alkaline-earth and ytterbium cold atomic gases, are highly non-trivial multicomponent quantum systems [3, 4]. Such systems extend beyond the physics of interacting spin-$\frac{1}{2}$ electrons found in condensed matter systems [5, 6]. They are very relevant both for high-precision measurement [7–9] and to enlarge the scope of cold atoms quantum simulators of many-body systems [10–13]. Additionally, atom-atom interactions can be made independent on the nuclear spin. This feature effectively enlarges the symmetry of the systems to the SU(*N*) one. Such a feature makes cold alkaline-earth atoms, especially with lattice confinements, an ideal platform to study exotic quantum matter, including higher spin magnetism, spin

liquids and topological matter [14–16] and, beyond condensed matter physics, in QCD [17].

Here, we consider $N_p$ SU(*N*) fermions with *repulsive* interaction, trapped in a ring-shaped circuit of mesoscopic size *L* [18] and pierced by an effective magnetic field. We study the persistent current response to this applied field, which provides a standard avenue to probe the coherence of the system [19].

Different regimes depending on the filling fraction $\nu = N_p/L$ are explored. *i)* For incommensurate $\nu$, the persistent current is non-vanishing for any value of the interaction. Monitoring the numerical results for the spectrum of the system with the exact Bethe ansatz analysis [20], we find that *as the effective magnetic flux increases, spinon excitations can be created in the ground state*. Such a remarkable phenomenon occurs as a specific 'screening' of the external flux, which being continuously adjustable quantity, can be compensated by spinons excitations (quantized in nature) only partially. This in turn results in an imbalance and causes the persistent current to display characteristic oscillations with a period of $1/N_p$ shorter than the bare flux quantum. For two-spin component fermions in the large interaction regime, such a phenomenon was studied in [21, 22]. We shall see that such a process depends on $N_p$, number of spin components and interaction in a non-trivial way. *ii)* In contrast with the SU(2) case [23], SU(*N*) fermions with $N > 2$ undergo a Mott quantum phase transition for a finite value of interaction $U = U_c$ at integer fillings. Accordingly, a metallic behaviour crosses over to a regime in which the current is exponentially suppressed. This regime is also monitored by Bethe ansatz [24] and corroborated by exact diagonalization and Density Matrix Renormalization Group (DMRG) [25, 26]. We shall see that, *despite the persistent current being mesoscopic in nature, the onset of the Mott transition is marked by a clear finite size scaling*. The onset to the gapped phase progressively hinders the spinon creation phenomenon.

*Methods* – A system of $N_p$ SU(*N*) fermions residing in a

ring-shaped lattice composed of $L$ sites threaded with a magnetic flux $\phi$ can be modeled using the Hubbard model [14]

$$\mathcal{H}_{\mathrm{SU}(N)} = -t \sum_{j=1}^{L} \sum_{\alpha=1}^{N} (e^{i\frac{2\pi\phi}{L}} c_{\alpha,j}^{\dagger} c_{\alpha,j+1} + \text{h.c.}) + \frac{U}{2} \sum_{j} n_j(n_j - 1)$$

(1)

where $c_{\alpha,j}^{\dagger}$ ($c_{\alpha,j}$) creates (annihilates) a fermion with colour $\alpha$, $n_j = \sum_{\alpha} c_{\alpha,j}^{\dagger} c_{\alpha,j}$ is the local particle number operator for site $j$. The parameters $t$ and $U > 0$ account for the hopping strength and on-site repulsive interaction respectively. The effective magnetic field is realized through Peierls substitution $t \to te^{i\frac{2\pi\phi}{L}}$. For $N = 2$, the Hubbard model describing spin-$\frac{1}{2}$ fermions is obtained. In this case, the Hamiltonian (1) is integrable by Bethe Ansatz (BA) for any $U/t$ and $\nu$ [23]. For $N > 2$, the BA integrability is preserved in the continuous limit of vanishing lattice spacing, (1) turning into the Gaudin-Yang-Sutherland model describing SU($N$) fermions with delta interaction [20, 27]; such a regime is achieved by (1) in the dilute limit of small $\nu$. Another integrable regime of (1) is obtained for $n_j = 1 \forall j$ and large repulsive values of $U \gg t$ for which the system is governed by the Lai-Sutherland model [14, 24].

In our approach, the exact solution plays a very important role in classifying the eigenstates of the system. According to the general theory of BA solvable models, the spectrum of the model is obtained through the solution of coupled transcendental equations, which are parameterized by a specific set of numbers called the quantum numbers [28]. Indeed, they label all the excitations. For the specific case of the integrable SU($N$) Hubbard models, the many-body excitations are labeled by quantum numbers: $I_a$, $a = 1 \ldots N_p$ and $J_{\beta_j}$, $\beta_j = 1 \ldots M_j$ for $j = 1 \ldots N - 1$ with $I_a$ and $J_{\beta_j}$ being integers or half-odd integers, and $M_j$ referring to the number of particles with a given component [20, 28, 29] (see Supplementary material). It is well known that, at zero flux $\phi$, the ground state is found to be characterized by $I_a = I_1, I_1 + 1, I_1 + 2, \ldots I_1 + N_p$ and $J_{\beta_1} = J_{\beta_1}, J_{\beta_1} + 1, J_{\beta_1} + 2, \ldots J_{\beta_1} + M_1$. Instead, sequences of $I_a = I_1 - 1, \vee, I_1 + 1, I_1 + 2, \ldots I_1 + N_p$ and $J_{\beta_j} = J_{\beta_j} - 1, \vee, J_{\beta_j} + 1, J_{\beta_j} + 2, \ldots J_{\beta_j} + M_j$ with 'holes' correspond to excitations; in particular holes in $\{J_\beta\}$ characterize the so-called spinon excitations [30]. For SU($N$) fermions, there can be $N - 1$ different types of such spinon states [31, 32]. For non-vanishing $\phi$, we shall see the quantum numbers configurations $\{I_a, J_\beta\}$ can change. For intermediate interactions and intermediate fillings, the model (1) is not integrable and approximated methods are needed to access its spectrum. Indeed, Hubbard models for SU(2) and SU($N$) fermions enjoy very different physics. For incommensurate fillings, a metallic behaviour is found with characteristic oscillations of the spin-spin and charge correlation functions that, for $N > 2$ can be coupled to each other. At integer $\nu$, fermions may be in a Mott phase. Such a phase is suppressed only exponentially for $N = 2$ [23]; in striking contrast, for

$N > 2$ the system displays a Mott transition for a finite value of $U/t$ [14, 33].

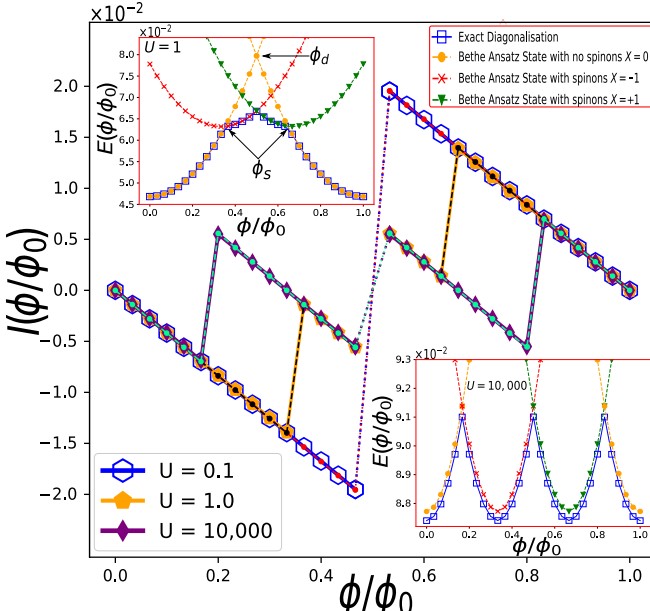

FIG. 1. Persistent current $I(\phi)$ at incommensurate filling for SU(3) fermions with different interaction strengths $U$ in the dilute filling regime of the Hubbard model. The exact diagonalization $L = 30, N_p = 3$ is monitored with the BA of the Sutherland-Gaudin-Yang model. The red, black and green dots in the main figure depict the Bethe ansatz results for the persistent current for $U = 0.1$, $1.0$ and $10,000$ respectively. These dots are meant to be a guide to the eye, to aid in perceiving the fractionalization of the persistent current with increasing interaction Insets show how the BA energies need to be characterized by $X \neq 0$, to be the actual ground state. At $U = 0$, the ground state energy is a periodic sequence of parabolas meeting at degeneracy points $\phi_d$ ($\phi_d = 1/2$ for the case in this figure). The values of the flux at which spinons are created $\phi_s$ have been included as an example in the top inset, which for an interaction $U = 1$, range from 0.37 to 0.63.

At mesoscopic size, the properties discussed above are displayed as specific traits [18]. We refer as mesoscopic effects as those ones arising on length scales that are comparable with the particles' coherence length. In this regime, even though the application of the magnetic flux does not change the nature of the possible excitations, we shall see that the latter may be indeed promoted to ground states. Our diagnostic tool is the persistent current, providing access to the particles' phase coherence [19, 34]. At zero temperature, the persistent current of the system is given by $I(\phi) = -\frac{\partial E_0}{\partial \phi}$ where $E_0$ is the ground state energy. The persistent current is of mesoscopic nature in that it corresponds to $1/L$ corrections of the ground state energy [19].

For a quantum system in a ring, the angular momentum is quantized (see [35, 36] for recent experiments). Accordingly, $I(\phi)$ displays a characteristic sawtooth behaviour, with a periodicity that Leggett proved to be fixed by the effective

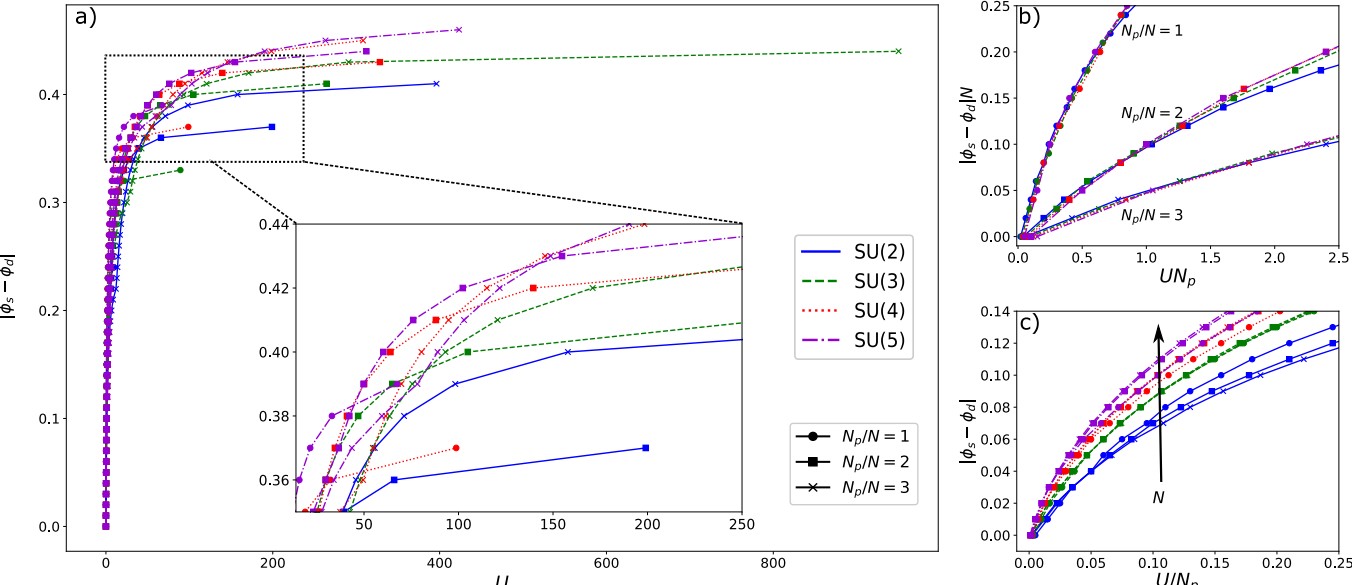

FIG. 2. Figures of merit for spinon creation in the ground state of SU($N$) fermions. We consider the minimum value of $U$ required for spinons to be created in the ground state for a given value of $\phi$; all the values of $\phi$ where a spinon is created are recorded. The displayed curves are calculated by monitoring all the distances $|\phi_s - \phi_d|$ at which the state with no spinons crosses states with any spinon states, where $\phi_s$ is the flux at which spinons are created and $\phi_d$ is the degeneracy point (see Fig. 1). **a)** Spinon creation flux distance $|\phi_s - \phi_d|$ against interaction $U$. The inset contains the data in the intermediate $U$ regime. **b)**Spinon creation flux distance against the interaction, rescaled by $N$ and $N_p$ respectively, in the limit of low $UN_p$. In this limit, spinon production is found to be a universal function of $N_p/N$ **c)** Spinon creation flux distance $|\phi_s - \phi_d|$ against interaction per particle $U/N_p$ in the low $U/N_p$ regime. One can observe the enhancement of spinon production with increasing $N$. All the presented results are obtained by BA of Gaudin-Yang-Sutherland model for $L = 40$, with $N_p = 1$(circles), 2(squares), 3(crosses) per spin component, with $N = 2, 3, 4, 5$. Thus, the dilute limit of the Hubbard model (1) is covered.

flux quantum $\phi_0$ of the system [37–39]. Furthermore, the persistent current is parity dependent: for systems with even (odd) number of spinless particles, the energy is decreased (increased) by the application of the external flux; therefore, the persistent current displays a paramagnetic (diamagnetic) behaviour. Leggett predictions are independent of disorder. In particular, the periodicity of the persistent current reflects the structure of the ground-state. For example in the case of a BCS ground-state, the period of the persistent current is halved due to the formation of Cooper pairs [37, 40]. Similarly for bosonic systems, the persistent current fractionalized with a period of $1/N_p$, indicating the formation of a bound state of $N_p$ particles [41]. The specific parity effects also hold for SU($N$) systems and SU(2) fermions [37, 38] (see Supplementary material).

In our approach, we combine exact diagonalization or DMRG analysis with, whenever possible, BA results. Specifically: in the integrable regimes of dilute systems (described by Gaudin-Yang-Sutherland model), and a filling of one-particle per site & large interactions (captured by the Sutherland model), the BA results (through the Bethe quantum numbers introduced above) are exploited as bookkeeping to monitor the eigenstates provided by the numerical results. This way, the nature and physical content of the system's ground state can be established as functions of the parameters. We shall see that the actual lowest energy of the system can only

be obtained with Bethe quantum numbers corresponding to spinon excitations. In the non-integrable regimes, we rely on numerical analysis. Here, only systems with an equal number of particles per species are considered. In the following, the energy scale is given by $t = 1$.

*Persistent current of SU(N) fermions at incommensurate fillings* – Our analysis begins in the low $\nu$ regimes (continuous limit) wherein we can rely on exact results based on the Gaudin-Yang-Sutherland model BA. The numerical analysis shows that, by increasing $\phi$, specific energy level crossings occur in the ground state of the system. The BA analysis (see Supplemental material) allows us to recognize such level crossings as ground state transitions between no-spinon states and spinon states. Specific $1/N_p$ periodic oscillations occur in the ground state energy as $\phi$ is varied; therefore, a curve with $N_p$ cusps/parabolic-wise segments per flux quantum emerges. Such a feature was evidenced for two-spin component fermions in the large interaction regime [21, 22]. *Here, we find that spinon creation defines a phenomenon occurring for any value of $U$; additionally, we shall see that the spinon creation mechanism displays a non-trivial dependence on the number of spin components $N$.* Indeed, the different $N - 1$ spinon configurations are found to play a relevant role for the phenomenon. The quantity $X = \sum_j^{N-1} \sum_{\beta_j}^{M_j} J_{\beta_j}$ can be exploited to characterize the properties of the specific spinon excitations that are created in the ground state.

Specifically, for small and intermediate $U$, while the system's ground state with no spinons is found to be non-degenerate, the one with spinons can be made of degenerate multiplets corresponding to Bethe states with distinct configurations of the quantum numbers $J_{\beta_j}$ (see inset of Fig. 1). By further increasing $U$, the spinon states organize themselves in multiplets of increasing degeneracy on a wider interval of the flux. At large but finite $U$, the exact BA analysis shows that the spectrum can be reproduced by a suitable continuous limit of a SU($N$) $t - J_{eff}$ model with $J_{eff} = 4E_\infty/(UL)$, where $E_\infty$ is the energy of the Gaudin-Yang-Sutherland model at infinite interaction (see Supplementary material). We remark that the specific features of the SU($N$) fermions enter the entire energy spectrum of the system through the SU($N$) quantum numbers $\{I_a, J_{\beta_j}\}$. In the limit of infinite $U$, the persistent current is analytically obtained as (derivations in Supplementary material)

$$I(\phi) = -2\left(\frac{2\pi}{L}\right)^2 \sum_a^{N_p} \left[I_a + \frac{X}{N_p} + \phi\right] \qquad (2)$$

Equation (2) shows that, in this regime, the persistent current displays $1/N_p$ reduced periodicity; such a phenomenon is observed for $UL/N_p \gg 1$, for any number of spin components $N$. Therefore, in this regime, *the bare flux quantum of the system is evenly shared among all the particles.* We note that, in the infinite $U$ regime, the ground state reaches the highest degeneracy (see inset of Fig.1).

As a global view of spinon creation in the ground state, we monitor, for different values of $U$, $N$, and $N_p$, the values of the flux $\phi_s$ at which the ground state energy in the system is no longer given by a state with no spinons – Fig 2. Such values provide the number of spinons that can be present in the ground state at a given $U$. *At moderate $U$, spinon production is found to be a universal function of the $N_p/N$–* see Fig. 2b; for systems with lower $N_p$, spinons are generated at a lower value of interaction. For large $U$, spinon production is dictated by $N_p$, with a fine structure that is determined by $N$: Systems with higher $N_p$ produce spinons at a lower value of $U$; for fixed $N_p$, systems with the lower value of $N_p/N$ generate spinons at a lower $U$– see inset of Fig 2a. Such a phenomenon depends on the specific degeneracies of the system discussed previously, that facilitate spinon creation by increasing $N$. This feature emerges also by analysing the dependence of the phenomenon on the interaction per particle $U/N_p$ – Fig. 2c. *We observe that $N$ enhances the spinon production .* While the number of spinons decreases with $N_p$ for $N = 2$, such a trend appears to be reversed for $N > 2$. For intermediate values of $U$, discontinuities arise in the curves in cases where $N_p/N > 1$ (see Supplementary material). These discontinuities correspond to jumps $\Delta X$ in the spinon character $X$. By comparing systems with the same $N_p$ but different $N$, we note that the discontinuities tend to be smoothed out by increasing $N$ and $L$ (see Supplementary material). The value of $\Delta X$ results to be parity dependent.

*Commensurate fillings regime* – At integer filling fractions $\nu = 1$, the system enters a Mott phase for $U > U_c$ (thermo-

dynamic limit). In this phase, a spectral gap opens. For small $U$, the current is a nearly perfect sawtooth. For our mesoscopic system, we observe that $I(\phi)$ is smoothed out, indicating the onset of the Mott phase transition by increasing $U$ (see Fig. 3a). Such a behavior is found to hold for all $N$. The gap indicating the onset of the Mott phase transition is studied in Fig. 3c (see Supplementary material). For $N = 2$ such gap opens at $U = 0$; for $N > 2$ the spectral gap opens at a finite value of $U$. Both the current amplitude $I_{max} = \max_\phi(I)$ and $\Delta E_{min}$ are suppressed exponentially for large $U$ – Fig. 3b and Fig. 3c. $\Delta E_{min}$ is around the same specific value $U$ for larger system sizes ($L \geq 8$), which depends on $N$ ($U \approx 2$ for $N = 3$, $U \approx 3$ for $N = 4$). We carry out a finite size scaling analysis [42] of the current $I$ for values of $U$ around the Mott instability. In Fig. 4a, the persistent currents display a crossing point at a particular value $U^* \approx 2.9$ (see also [43–46]); a clear data collapse is obtained in Fig. 4b.

The onset to a gapped phase affects the spinon creation process substantially. For $N = 2$ ($U^* = 0$), spinon states have energies larger than the ground state energy for any value of $U$. In contrast for $N > 2$, spinons can be created for $U < U^*$ (see inset of Fig. 3a); for $U > U^*$ spinon energies result to be well separated from the ground state energy. We note that the Lai-Sutherland BA results can reproduce the qualitative features of the low lying states of the model even for intermediate $U$ obtained by numerics; as expected, for large $U$, BA and numerics match exactly (see Supplementary material).

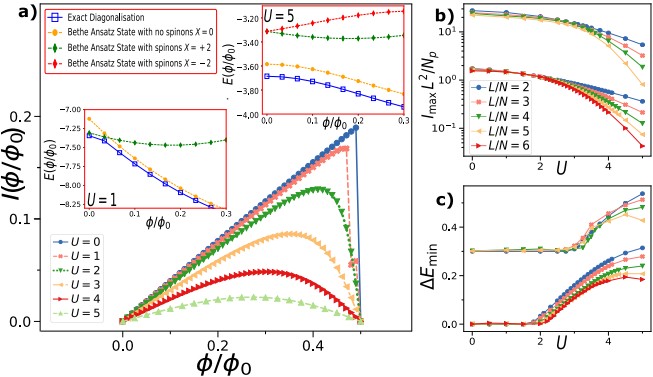

FIG. 3. SU($N$) persistent current $I(\phi)$ at integer filling. **a)** $I(\phi)$ for $N = 3$, $L = 9$ against flux $\phi$. Insets display BA results of the Sutherland-Lai model for different spinon configurations $X$ for $N = 3$ and $N_p = L = 6$ compared with exact diagonalization. The ground state energy at small $\phi$ can only be reached with spinon Bethe quantum numbers configuration. **b)** Maximal current $I_{max} = \max_\phi(I)$ for $N = 3$ (lower curves) and $N = 4$ (upper curves, shifted by factor 20) plotted against $U$ **c)** Minimal energy gap $E_{min}$ against $U$ for $N = 3$ (lower curves) and $N = 4$ (upper curves, shifted by 0.3). All curves with $L > 9$ were calculated with DMRG.

*Conclusions*– In this work, the coherence of a quantum gas of SU($N$) interacting fermions as quantified by the persistent current, is studied. The analysis is carried out both for incommensurate and commensurate filling $\nu$ regimes. We highlight the nature of the ground state of the system by corroborating the numerical analysis (exact diagonalization and DMRG)

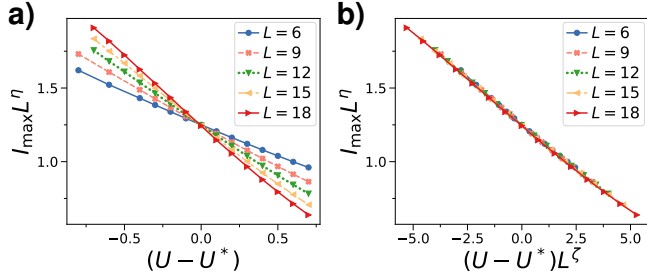

FIG. 4. Finite size scaling of the persistent current for $N = 3$. **a)** Finite size critical crossing of $I_{max}$ at $U^* = 2.9$ **b)** Data collapse. $L = 6, 9$ were calculated with exact diagonalization; larger $L$ were obtained with DMRG. The critical indices are $\eta \approx 0.2$ and $\zeta \approx 0.7$

with Bethe ansatz, which allows the access to the specific physical nature of the system's states. For both incommensurate and commensurate $\nu$, the ground state can have spinon nature. Such a remarkable phenomenon implies that *the spin correlations can lead to a re-definition of the system's effective flux quantum* and, for incommensurate $\nu$ cases, yields the $1/N_p$ fractional periodicity for the persistent current observed at large $U$ (see insets of Fig.1). The reduction of the effective flux quantum indicates that a form of 'attraction from repulsion' can occur in the system, with this feature being consistent with [47]. Despite the similarities, such a phenomenon follows a very different route from the flux quantum fractionalization occurring for electrons with pairing force interaction (that could compared to our study for $N = 2$ only) [37, 38] and for bosons with attractive interaction (occurring as consequence of quantum bright solitons formation) [48, 49]: *For SU(N) fermions the persistent current and the aforementioned redefinition of the flux quantum reflects the coupling between the spin and matter degrees of freedom.* The ground state spinon creation displays a marked dependence on the number of spin components $N$ with distinctions between the $N = 2$ and $N > 2$ cases (see Fig. 2).

*At moderate $U$, spinon production is found to be a universal function of $N_p/N$ – see Fig. 2b.* For integer $\nu$, spinon creation is suppressed by increasing $U$. The sawtooth shape of the current is smoothed out (see Fig. 3). This feature arises since the Mott gap hinders both the motion of the particles and the creation of spinons in the ground state. Remarkably, *a clear finite size scaling behaviour is observed for $N > 2$, albeit the persistent current is a mesoscopic quantity (see Fig. 4)*. Such a result provides an operative route for the detection of the Mott phase transition in SU(N) systems, a notoriously challenging problem in the field.

We believe that systems in physical conditions and parameter ranges as discussed here, can be realized experimentally on several physical platforms, including cold atom quantum technology [5, 6, 50, 51] with the twist provided by atomtronics [2]. Recently, the persistent current of SU(2) fermions was experimentally realized [52]. The momentum distribution through the time of flight expansion of cold atom systems has been demonstrated to provide a precise probe for

persistent currents [2]. Most of the features that we observe in the persistent current are expected to emerge in time of flight images of the system. Finite temperature is expected to reduce the visibility of the time of flight images. Nonetheless, based on the predictions for $N = 2$, the main features of the persistent current are expected to be feasibly detectable [53]. The temperature can activate transitions between different angular momentum states, thereby causing a decay in the amplitude of the persistent current [54]. Recently, the dependence of persistent currents on finite temperature effects has been investigated for SU(2) fermions [55]. However, the actual decay of the persistent current of SU(N) fermions needs an in-depth study.

In particular, here we mention that it was previously demonstrated how the flux fractionalization could allow to approach the Heisenberg quantum limit for rotation sensing [49]. Our study indicates how SU(N) systems can provide the platform for high precision sensors.

*Acknowledgements* We thank Anna Minguzzi for discussions. The Grenoble LANEF framework ANR-10-LABX-51-01 are acknowledged for their support with mutualized infrastructure.

* On leave from Dipartimento di Fisica e Astronomia 'Ettore Majorana', Università di Catania, Italy

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

# Supplementary material
*Persistent Current of SU(N) Fermions.*

In the following sections, we provide supporting details of the theory discussed in the main manuscript.

The derivation of the persistent current for SU(N) fermions is sketched out in the limit of infinite interaction $U$. The analytics are carried out for the two integrable limits of the Hubbard model: incommensurate low filling fractions and integer fillings. A specific analysis is devoted to the energy and consequently the persistent current at large but finite $U$. The persistent current undergoes a non-trivial change of the bare flux quantum. This feature occurs because of the presence of spinons in the ground state of the system. Spinons of different types correspond to specific Bethe quantum numbers configurations. The Bethe quantum number configurations needed for the given value of $X$ are provided. We then proceed to discuss $|\phi_s - \phi_d|$ providing the figure of merit for the generation of spinons. Spinon generation is inhibited for commensurate fillings due to a spectral gap that opens up for a finite value of $U$. Lastly, the parity effect for incommensurate systems is considered.

## Derivation of the Persistent Current in the limit of infinite $U$ for SU(N) Fermions

The derivation of the persistent current for SU(N) fermions in the limit of infinite interaction $U$, is sketched out for the two integrable limits of the SU(N) Hubbard model.

A system of interacting fermions with SU(N) spin symmetry residing on a chain of length $L$ threaded by an effective magnetic flux $\phi$, is described by the Gaudin-Yang-Sutherland model [20, 27],

$$\mathcal{H} = \sum_{m=1}^{N} \sum_{i=1}^{N_m} \left( -\iota \frac{\partial}{\partial x_{i,m}} - \frac{2\pi}{L}\phi \right)^2 + 4U \sum_{i<j,m,n} \delta(x_{i,m} - x_{j,n}) \tag{3}$$

where $N_m$ is the number of electrons with colour $\alpha$ of SU(N) symmetry with $m = 1, \ldots N$. The model is integrable by Bethe ansatz and is given by the following set of equations.

$$e^{\iota(k_j L - \phi)} = \prod_{\alpha=1}^{M_1} \frac{4\left(k_j - \lambda_\alpha^{(1)}\right) + \iota U}{4\left(k_j - \lambda_\alpha^{(1)}\right) - \iota U} \qquad j = 1, \ldots, N_p \tag{4}$$

$$\prod_{\beta \neq \alpha}^{M_r} \frac{2\left(\lambda_\alpha^{(r)} - \lambda_\beta^{(r)}\right) + \iota U}{2\left(\lambda_\alpha^{(r)} - \lambda_\beta^{(r)}\right) - \iota U} = \prod_{\beta=1}^{M_{r-1}} \frac{4\left(\lambda_\alpha^{(r)} - \lambda_\beta^{(r-1)}\right) + \iota U}{4\left(\lambda_\alpha^{(r)} - \lambda_\beta^{(r-1)}\right) - \iota U} \cdot \prod_{\beta=1}^{M_{r+1}} \frac{4\left(\lambda_\alpha^{(r)} - \lambda_\beta^{(r+1)}\right) + \iota U}{4\left(\lambda_\alpha^{(r)} - \lambda_\beta^{(r+1)}\right) - \iota U} \qquad \alpha = 1, \ldots, M_r \tag{5}$$

for $r = 1, \ldots, N-1$ where $M_0 = N_p$, $M_N = 0$ and $\lambda_\beta^{(0)} = k_\beta$. $N_p$ denotes the number of particles, $M_r$ corresponds to the colour with $k_j$ and $\lambda_\alpha^{(r)}$ being the charge and spin momenta respectively. The energy corresponding to the state for every solution of these equations is $E = \sum_j^{N_p} k_j^2$.

Taking the SU(3) case as an example, one obtains a set consisting of three non-linear equations

$$e^{\iota k_j L} = \prod_{\alpha=1}^{M_1} \frac{4(k_j - \lambda_\alpha^{(1)}) + \iota U}{4(k_j - \lambda_\alpha^{(1)}) - \iota U} \tag{6}$$

$$\prod_{\beta \neq \alpha}^{M_1} \frac{2(\lambda_\alpha^{(1)} - \lambda_\beta^{(1)}) + \iota U}{2(\lambda_\alpha^{(1)} - \lambda_\beta^{(1)}) - \iota U} = \prod_{\beta=1}^{M_0 = N_p} \frac{4(\lambda_\alpha^{(1)} - k_\beta) + \iota U}{4(\lambda_\alpha^{(1)} - k_\beta) - \iota U} \prod_{\beta=1}^{M_2} \frac{4(\lambda_\alpha^{(1)} - \lambda_\beta^{(2)}) + \iota U}{4(\lambda_\alpha^{(1)} - \lambda_\beta^{(2)}) - \iota U} \tag{7}$$

$$\prod_{\beta \neq \alpha}^{M_2} \frac{2(\lambda_\alpha^{(2)} - \lambda_\beta^{(2)}) + \iota U}{2(\lambda_\alpha^{(2)} - \lambda_\beta^{(2)}) - \iota U} = \prod_{\beta=1}^{M_1} \frac{4(\lambda_\alpha^{(2)} - \lambda_\beta^{(1)}) + \iota U}{4(\lambda_\alpha^{(2)} - \lambda_\beta^{(1)}) - \iota U} \tag{8}$$

which can be re-written in logarithmic form as

$$k_j L + 2 \sum_{\alpha=1}^{M_1} \arctan\left(\frac{4(k_j - \lambda_\alpha^{(1)})}{U}\right) = 2\pi(I_j + \phi) \quad j = 1, \ldots, N_p \tag{9}$$

$$2 \sum_{\beta=1}^{N_p} \arctan\left(\frac{4(\lambda_\alpha^{(1)} - k_\beta)}{U}\right) + 2 \sum_{a=1}^{M_2} \arctan\left(\frac{4(\lambda_\alpha^{(1)} - l_a)}{U}\right) - 2 \sum_{\beta=1}^{M_1} \arctan\left(\frac{2(\lambda_\alpha^{(1)} - \lambda_\beta^{(1)})}{U}\right) = 2\pi J_\alpha \quad \alpha = 1, \ldots, M_1 \tag{10}$$

$$2 \sum_{\beta=1}^{M_1} \arctan\left(\frac{4(l_a - \lambda_\beta^{(1)})}{U}\right) - 2 \sum_{b=1}^{M_2} \arctan\left(\frac{2(l_a - l_b)}{U}\right) = 2\pi L_a \quad a = 1, \ldots, M_2 \tag{11}$$

where $\lambda_\beta^{(2)}$ was changed to $l_a$ for the sake of convenience with $I_j$, $J_\alpha$ and $L_a$ being the Bethe quantum numbers, the first being associated with charge momenta and the other two for spin momenta. Carrying out a summation over $\alpha$ and over $a$ for Equations (10) and (11) respectively,

$$2 \sum_{\alpha=1}^{M_1} \sum_{\beta=1}^{N_p} \arctan\left(\frac{4(\lambda_\alpha^{(1)} - k_\beta)}{U}\right) + 2 \sum_{\alpha=1}^{M_1} \sum_{a=1}^{M_2} \arctan\left(\frac{4(\lambda_\alpha^{(1)} - l_a)}{U}\right) - 2 \sum_{\alpha=1}^{M_1} \sum_{\beta=1}^{M_1} \arctan\left(\frac{2(\lambda_\alpha^{(1)} - \lambda_\beta^{(1)})}{U}\right) = 2\pi \sum_{\alpha=1}^{M_1} J_\alpha \tag{12}$$

$$2 \sum_{a=1}^{M_2} \sum_{\beta=1}^{M_1} \arctan\left(\frac{4(l_a - \lambda_\beta^{(1)})}{U}\right) - 2 \sum_{a=1}^{M_2} \sum_{b=1}^{M_2} \arctan\left(\frac{2(l_a - l_b)}{U}\right) = 2\pi \sum_{a=1}^{M_2} L_a \tag{13}$$

and noting that the last term on the left hand side in both of the above equations goes to zero, leads one to the following expression

$$2 \sum_{\alpha=1}^{M_1} \sum_{\beta=1}^{N_p} \arctan\left(\frac{4(\lambda_\alpha^{(1)} - k_\beta)}{U}\right) = 2\pi\left(\sum_{\alpha=1}^{M_1} J_\alpha + \sum_{a=1}^{M_2} L_a\right) \tag{14}$$

In the limit $\frac{U}{N_p} \to \infty$, the $k_j$ terms can be neglected since they are significantly smaller in magnitude compared to the spin momenta. Consequently,

$$2 \sum_{\alpha=1}^{M_1} \sum_{\beta=1}^{N_p} \arctan\left(\frac{4\lambda_\alpha^{(1)}}{U}\right) = 2\pi\left(\sum_{\alpha=1}^{M_1} J_\alpha + \sum_{a=1}^{M_1} L_a\right) \implies 2 \sum_{\alpha=1}^{M_1} \arctan\left(\frac{4\lambda_\alpha^{(1)}}{U}\right) = \frac{2\pi}{N_p}\left(\sum_{\alpha=1}^{M_1} J_\alpha + \sum_{a=1}^{M_1} L_a\right) \tag{15}$$

which upon substitution in Equation (9) yields

$$k_j L = 2\pi\left[I_j + \frac{1}{N_p}\left(\sum_{\alpha=1}^{M_1} J_\alpha + \sum_{a=1}^{M_2} L_a\right) + \phi\right] \tag{16}$$

Squaring the above expression,

$$k_j^2 = \left(\frac{2\pi}{L}\right)^2\left[I_j^2 + 2I_j\left(\frac{X}{N_p} + \phi\right) + \left(\frac{X}{N_p}\right)^2 + 2\phi\frac{X}{N_p} + \phi^2\right] \tag{17}$$

the ground state energy of the system is given by

$$E_0 = \sum_j^{N_p} k_j^2 = \left(\frac{2\pi}{L}\right)^2\left[\sum_j^{N_p} I_j^2 + 2 \sum_j^{N_p} I_j\left(\frac{X}{N_p} + \phi\right) + N_p\left(\frac{X}{N_p}\right)^2 + N_p\left(2\phi\frac{X}{N_p} + \phi^2\right)\right] \tag{18}$$

assuming the $I_j$ quantum numbers are a consecutive integer/half-integer set, where $X = \left(\sum_{\alpha=1}^{M_1} J_\alpha + \sum_{a=1}^{M_2} L_a\right)$. At zero temperature the persistent current of the system is defined as,

$$I(\phi) = -\frac{\partial E_0}{\partial \phi} \tag{19}$$

Therefore, the persistent current in the limit of infinite $U$ turns out to be,

$$I(\phi) = -2\left(\frac{2\pi}{L}\right)^2 \sum_{j}^{N_p} \left[I_j + \frac{X}{N_p} + \phi\right] \tag{20}$$

In the case of SU($N$) fermions, one would still have the same expression for the persistent current. The only difference is that $X = \sum_{j}^{N-1} \sum^{\alpha_j} J_{\alpha_j}$.

The other integrable limit of the SU($N$) Hubbard model, is for commensurate filling fractions in the presence of a lattice. The model is described by the Lai-Sutherland model [14, 24] with the energy of the system being given by $E = -2 \sum_{j}^{N_p} \cos k_j$. The Bethe ansatz equations for this model are similar to the ones outlined in Equations (4) and (5). However, in this case in Equation (4) there is $\sin k_j$ instead of $k_j$ on the right hand side and for Equation (5) one substitutes $\lambda_{\beta}^{(0)} = \sin k_{\beta}$ when required. By following the same procedure one arrives to Equation (16). Substituting this expression in the energy of the system, one arrives to

$$E_0(\phi) = -E_m \cos\left[\frac{2\pi}{L}\left(D + \frac{X}{N_p} + \phi\right)\right] \tag{21}$$

and in turn the persistent current is of the following form,

$$I(\phi) = -E_m\left(\frac{2\pi}{L}\right)\sin\left[\frac{2\pi}{L}\left(D + \frac{X}{N_p} + \phi\right)\right] \tag{22}$$

where $E_m = 2\frac{\sin\left(\frac{N_p\pi}{L}\right)}{\sin\left(\frac{\pi}{L}\right)}$ where $D = \frac{I_{max}+I_{min}}{2}$, which comes about due to the $I_j$ being consecutive for the ground state configuration. The above expression is a generalization of the ground state energy for SU(2) fermions obtained in [21, 22]. In particular, at infinite $U$ for the same number of particles the pre-factor $E_m$ is the same for SU($N$) as it was for SU(2). This in turn implies that the ground state energy for fermions carrying different SU($N$) spin, say SU(2) and SU(3), will coincide if their phase shift is the same. The same also holds true for expression (18).

## CORRECTIONS TO THE INFINITE $U$ LIMIT: DERIVATION OF THE ENERGY SPIN CORRECTION

In this section we generalize the energy spin correction, obtained for SU(2) fermions in [22], for SU($N$) fermions. At infinite $U$ the system is highly degenerate, meaning that there are multiple ways of choosing the spin rapidity $J_\alpha$ distribution [21, 22]. In order to find out the lowest energy state at finite $U$ when the degeneracy is lifted, leading order $\frac{1}{U}$ corrections have to be introduced for the Bethe ansatz equations at infinite $U$. When $U$ is at infinity, the charge momenta $k_j$ are of order unity whilst the spin momenta $\lambda_\beta$ are of order $U$. With this picture in mind, we expand the arctangent function in Equation (9) to leading order in $\frac{k_j}{U}$. Defining the scaled variables $x_\alpha$ as,

$$x_\alpha = \lim_{U\to\infty}\left(\frac{2\lambda_\alpha}{U}\right) \tag{23}$$

through Taylor expansion one finds that,

$$f(x + h) = \arctan(2x_\alpha) - 2\frac{k_j}{U}\frac{1}{x_\alpha^2 + \frac{1}{4}} \tag{24}$$

Therefore, for large but finite $U$, the $k_j$ have leading $\frac{1}{U}$ corrections,

$$\delta k_j = -2\frac{k_j}{UL}\sum_{\alpha}^{M}\frac{1}{x_\alpha^2 + \frac{1}{4}} \tag{25}$$

where the $x_\alpha$ has to satisfy the remaining Bethe equations which in the SU(3) case for example are Equations (11) and (12). The total ground state energy reads,

$$E = \sum_{j}^{N_p}(k_j + \delta k_j)^2 = \sum_{j=1}^{N_p}\left(k_j^2 + 2k_j\delta k_j + (\delta k_j)^2\right) \tag{26}$$

Therefore, the leading order $\frac{1}{U}$ correction is given by

$$+ 2k_j \delta k_j = -\frac{4}{UL} \sum_j^{N_p} k_j^2 \sum_\alpha^M \frac{1}{x_\alpha^2 + \frac{1}{4}} = J_{eff} \sum_\alpha^M \frac{1}{x_\alpha^2 + \frac{1}{4}} \tag{27}$$

In the presence of a lattice, the energy correction is of a similar form

$$+ 2\delta k_j \sin(k_j) = J_{eff} \sum_\alpha^M \frac{1}{x_\alpha^2 + \frac{1}{4}} \tag{28}$$

but $J_{eff} = -\frac{4}{UL}\left( \sum_{j=1}^{N_p} \sin^2 k_j \right)$ in this case, whereby $k_j$ in Equation (27) was replaced by $\sin k_j$. The leading order $\frac{1}{U}$ correction to the Bethe ansatz equations for SU($N$) fermions has the same expression as the one obtained for SU(2) in [22]. This was to be expected since Equation (9), which is the primary equation relating the charge and spin rapidities, is the same for all SU($N$).

## BETHE ANSATZ SPINON CONFIGURATIONS

To obtain the minimum energy for a given value of the flux $\phi$, one requires that the summation over the spin rapidities satisfies the degeneracy point equation having the form [21, 22]

$$\frac{2w - 1}{2N_p} \le \phi + D \le \frac{2w + 1}{2N_p} \quad \text{where} \quad X = -w \tag{29}$$

with $w$ only being allowed to have integer or half-integer values due to the nature of the spin rapidities.

Consider the case of three fermions with SU(3) spin. There are three sets of quantum numbers: one pertaining to the charge momenta $I_j$ and the other two belonging to the spin momenta denoted as $J_{\alpha_1}$ and $J_{\alpha_2}$. The ground state configuration for such a system is given as $I_j = \{-1, 0, 1\}$, $J_{\alpha_1} = \{-0.5, 0.5\}$ and $J_{\alpha_2} = \{0\}$. The correction of the spin quantum numbers for all the values of the flux per Equation (27) is as follows

| Magnetic flux | $J_{\alpha_1}$ | $J_{\alpha_2}$ | $X$ |
|---|---|---|---|
| $0.0 - 0.1$ | $\{-0.5, 0.5\}$ | $\{0\}$ | $0$ |
| $0.2 - 0.5$ | $\{-1.5, 0.5\}$ | $\{0\}$ | $-1$ |
| $0.6 - 0.8$ | $\{-0.5, 1.5\}$ | $\{0\}$ | $+1$ |
| $0.9 - 1.0$ | $\{-0.5, 0.5\}$ | $\{0\}$ | $0$ |

TABLE I. Spin quantum number configurations with the flux for $N_p = 3$ with SU(3) spin with $M_1 = 2$ and $M_2 = 1$.

As can be seen from Table (I), in cases where $X = 0$, the spin quantum number configuration is different from the ground state one and 'holes' are introduced such that the spin quantum number configurations are no longer consecutive, with the $I_j$ set remaining unchanged. There are two notable points worthy of mention. The first is that one could have chosen a different way to arrange the set of quantum numbers. An alternative arrangement is given by Table (II). The target value $X$ is reached via a different configuration, which in turn leads to a degenerate state. Such a phenomenon is a characteristic property of SU($N$) systems that is not present for SU(2). As $N$ increases, the number of degenerate states that are present in the system increases due to the various Bethe quantum number configurations that one can adopt.

| Magnetic flux | $J_{\alpha_1}$ | $J_{\alpha_2}$ | $X$ |
|---|---|---|---|
| $0.0 - 0.1$ | $\{-0.5, 0.5\}$ | $\{0\}$ | $0$ |
| $0.2 - 0.5$ | $\{-0.5, 0.5\}$ | $\{-1\}$ | $-1$ |
| $0.6 - 0.8$ | $\{-0.5, 0.5\}$ | $\{+1\}$ | $+1$ |
| $0.9 - 1.0$ | $\{-0.5, 0.5\}$ | $\{0\}$ | $0$ |

TABLE II. Alternative spin quantum number configurations with the flux for $N_p = 3$ with SU(3) spin with $M_1 = 2$ and $M_2 = 1$.

The other point concerns the value of $X$ for $\phi = 0.6 - 0.8$ and $\phi = 0.9 - 1.0$. According to Equation (29), $X$ should be equal to $-2$ and $-3$ respectively. The reason behind this is due to the fact that the degeneracy equation has to be applied within a specific

flux range that depends on the parity of the system: for a flux in the interval of $-0.5$ to $0.5$ for $N_p = N(2n + 1)$ and in the range of $\phi = 0.0$ to $1.0$ in the case of $N_p = N(2n)$. The ground state energy of the system is given by a series of parabolas in the absence of an effective magnetic flux. These parabolas each have a well defined angular momentum $l$. They intersect at the degnercy points, which is parity dependent, and are shifted with respect to each other by a Galilean translation [56]. Consequently, when the magnetic flux piercing the system falls outside the range outlined previously, one needs to change the $I_j$ quantum numbers in order to offset the increase in angular momentum $l$ that one obtains on going to the next energy parabola.

For positive $\phi$ one requires that the $I_j$ quantum numbers need to all be shifted by one to the left. For example in the case considered above for $\phi > 0.5$, the $I_j$ go from $\{-1, 0, 1\}$ to $\{-2, -1, 0\}$ for $0.5 < \phi < 1.5$. On going to the next parabola, they would need to be shifted again by one to the left. In the case of negative $\phi$, the shift occurs to the right.

Note that there are other combinations of the quantum numbers, not outlined in Tables I amd II, whose total sum reaches the target value of $X$. However, these configurations do not give the lowest value for the energy as the ones mentioned, even though the value of $X$ is the same. At infinite $U$, the system is solely dependent on the value of $X$ and not on the arrangement of the spin quantum number configuration. Consequently, the system is highly degenerate. This is also observed in the SU(2) case. However, as mentioned in the derivation of the energy correction, the degeneracy is lifted on going to large but finite $U$ and one is left with only one combination that gives the lowest energy in the case of SU(2) systems. On the other hand, for SU($N$) systems whilst this degeneracy is also lifted, they also benefit from an extra 'source' of degeneracy due to the different configurations of the Bethe quantum numbers as shown in Tables I and II.

## SPINON CREATION IN THE GROUND STATE FOR SU($N$) FERMIONS

The SU($N$) Hubbard model is not integrable in all limits, unlike its SU(2) counterpart. The Hamiltonian is integrable by Bethe ansatz for incommensurate and commensurate filling fractions. In this section, we take a look at spinon creation for SU($N$) fermions in these two regimes.

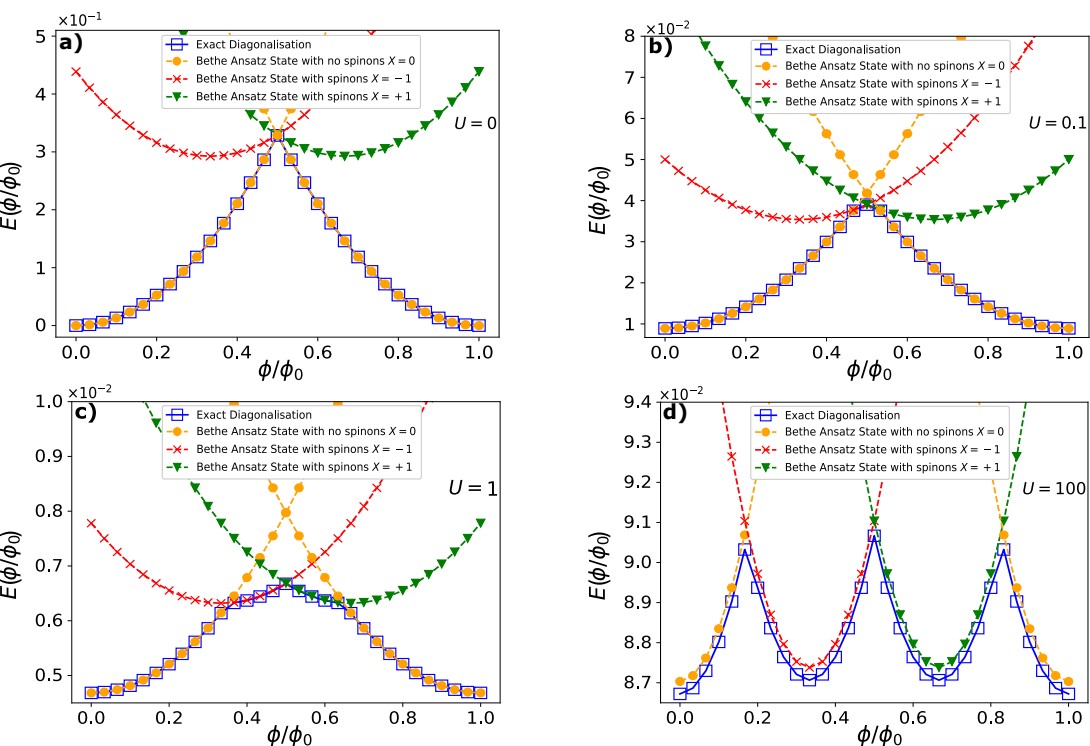

FIG. 5. Spinon creation in incommensurate SU($N$) fermionic systems. The case of $N = 3$ is considered for $N_p = 3$ fermions residing on a ring composed of $L = 30$ sites. The above figures show how the Bethe ansatz energies need to be characterized by spinon quantum numbers in order to have the actual ground state for various values of the interaction $U$. All curves are calculated with the Bethe ansatz of the Gaudin-Yang-Sutherland model and exact diagonalization.

For a system with incommensurate filling fractions, spinons are created with increasing $U$ as can be observed from Fig. 5. Level crossings occur between the ground state with no spinons and levels with spinon character $X$, with the value of $X$ obtained as outlined in the previous section. The creation of spinons starts out around the degeneracy point $\phi_d$ (see Fig. 5b), which is parity dependent. The degeneracy point $\phi_d$ is 0 for $N_p = N(2n)$ and 0.5 for $N_p = N(2n + 1)$ systems. Comparing Fig. 5a and Fig. 5d, we observe that the elementary flux quantum $\phi_0$ has been renormalized in the latter case and that $1/N_p$ periodicity is achieved, resulting in $N_p$ cusps/parabolic-wise segments that corresponds to 3 in this case.

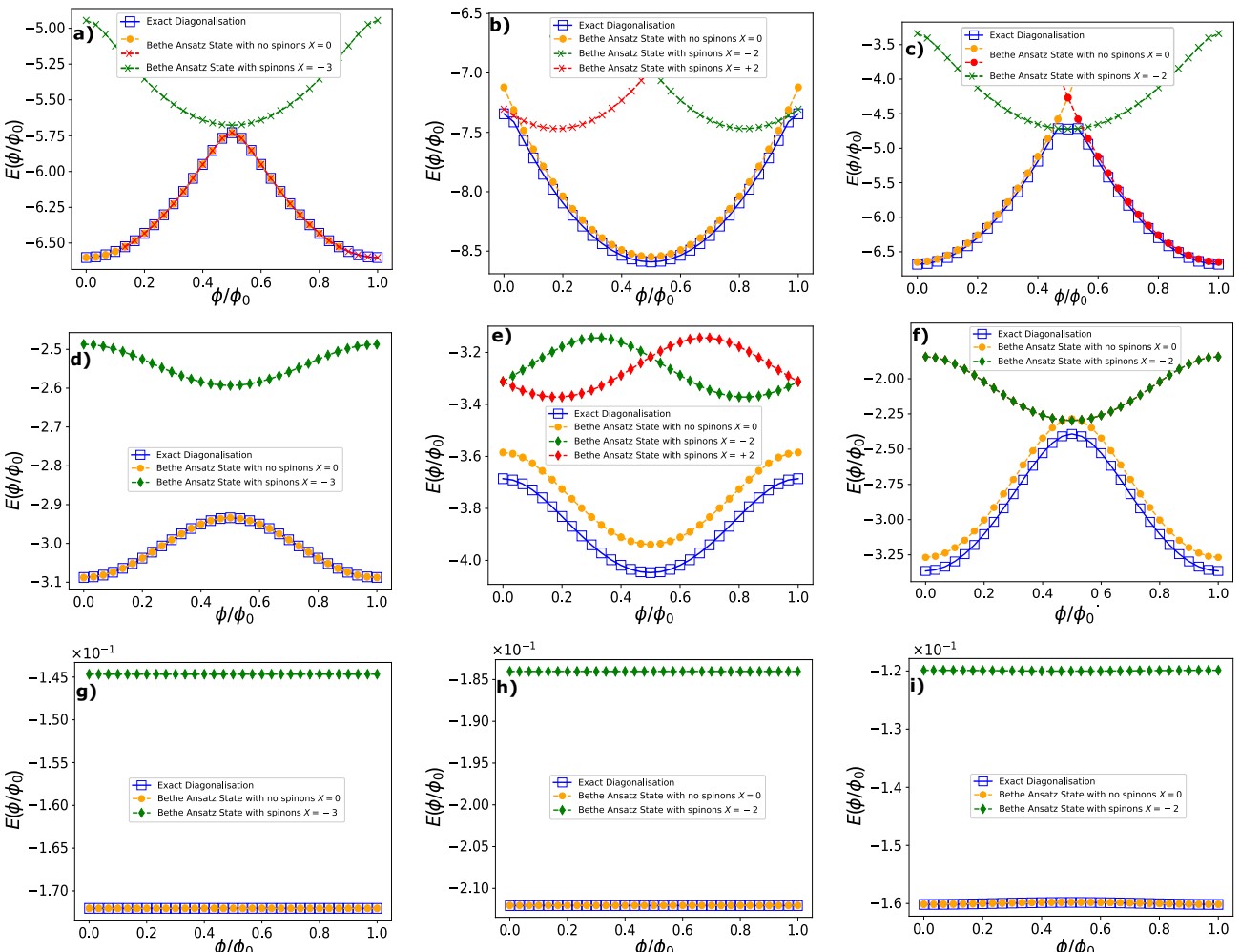

FIG. 6. Spinon creation in commensurate SU($N$) fermionic systems. The systems taken in consideration are SU(2) with $N_p = 6$ (left column), SU(3) with $N_p = 6$ (middle column) and SU(4) with $N_p = 4$ (right column). The different Bethe ansatz energies of the Lai-Sutherland model characterized by different spinon configurations needed to make up the ground state of the system are considered for different values of interaction with $U = 1$ (top row), $U = 5$ (middle row) and $U = 100$ (last row). All the presented results are obtained with Bethe ansatz of the Gaudin-Yang-Sutherland model for $N_p = L$. The Bethe ansatz states with no spinons, having two different colours (orange and red), are used to indicate that the $I_j$ quantum numbers are shifted due to being in different energy parabolas. In the case of $U = 1$, the Bethe ansatz did not converge for certain values of the flux. This does not have an impact on what we are trying to discuss here and so they were left out.

In the case of commensurate filling fractions, spinon creation is drastically impacted by a spectral gap that opens around the transition to the Mott phase (see Fig. 7). The energy gap is determined as the minimal gap for any flux $\Delta E = \min_\phi(\Delta E)$. For the special case $N = 2$, the gap opens at $U = 0$, whereas for any other $N$ it opens at non-zero $U$ indicating the onset to the Mott phase transition. Indeed if spinon creation in a system with SU(2) fermions (Figs. 6a),d),g)) is compared to systems with SU(3) (Figs. 6b),e),h)) and SU(4) (Figs. 6c),f),i)) spin components, we note that no spinons are created in the SU(2) case for any value of $U$. On the other hand for SU($N$) systems, spinon creation is present in the system for values of $U$ below the threshold value of where the transition happens $U^*$, which was calculated to be around 2.9. An interesting feature that pops up, is that after passing $U^*$, one no longer needs to change the $I_j$ quantum numbers on going from one energy parabola to the other, as can be seen by comparing (Figs. 6c),f)).

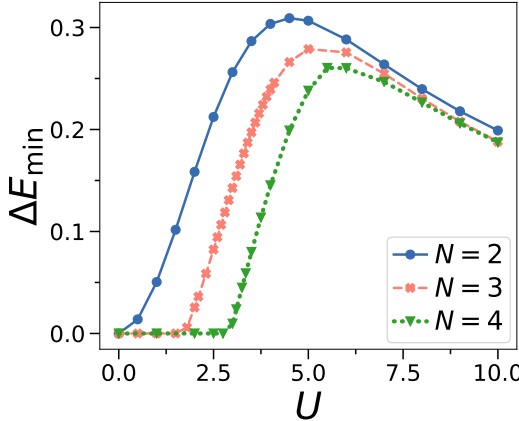

FIG. 7. SU($N$) energy gap at integer filling. Minimal energy gap $E_{\min} = \min_\phi(\Delta E)$ for different $N$ against $U$ at comparable system sizes ($N = 2$ and $N = 4$ with $L = 8$ and $N = 3$ with $L = 9$). All curves were obtained by exact diagonalization.

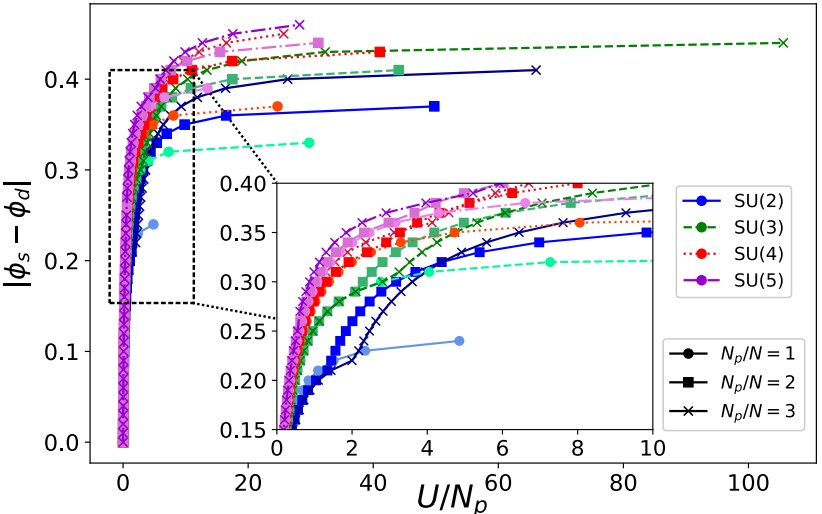

FIG. 8. Figure of merit for spinon creation in the ground state of SU($N$) fermions against the interaction per particle $U/N_p$. We consider the minimum value of $U$ required for spinons to be created in the ground state for a given value of $\phi$; all the values of $\phi$ where a spinon is created are recorded. The displayed curves are calculated by monitoring all the distances $|\phi_s - \phi_d|$ at which the state with no spinons crosses states with any spinon configurations where $\phi_s$ is the flux at which spinons are created and $\phi_d$ is the degeneracy point. The inset contains the data in the intermediate regime. The discontinuities observed in the intermediate $U/N_p$ regime when $N_p/N > 1$, are more pronounced for larger values of $N_p/N$ for a system with the same $N_p$ but different $N$. All the presented results are obtained by Bethe ansatz of Gaudin-Yang-Sutherland model for $L = 40$, with $N_p = 1$(circles), 2(squares), 3(crosses) per spin component, with $N = 2, 3, 4, 5$.

A good way to visualize the impact of SU($N$) fermions on spinon creation in the ground state is by having a figure of merit. By considering all the values of the flux $\phi$ at which spinons can be created and noting down the value of the interaction $U$, at which there is a level crossing between the ground state and an excited state, one can obtain this figure of merit. The curves displayed in Fig. 8 are obtained by monitoring the spinon flux distance $|\phi_s - \phi_d|$ at which there are crossings between states with spinons and those without, with $\phi_s$ being the flux at which spinons are created and $\phi_d$ being the degeneracy point. Figure depicts the spinon creation flux distance against the interaction per particle $U/N_p$. The inset denotes the data in the intermediate $U/N_p$ regime. Due to the specific $N - 1$ types of excitations that are inherently present in SU($N$) fermions for $N > 2$, spinon creation is facilitated with $N$. This can be clearly seen from Figs. 9a),b). In the case where $N_p/N > 1$, discontinuities arise in the intermediate $U$ regime as can be seen from the insets of Figs. 9a),b). The discontinuites arise due to jumps $\Delta X$ in the spinon character $X$ and are absent when $N_p/N = 1$ (inset of Fig. 9a). Additionally, when comparing systems containing the same $N_p$, the discontinuities tend to smoothen out with increasing $N$ and $L$. This can be clearly seen from Figs. 9b),c)

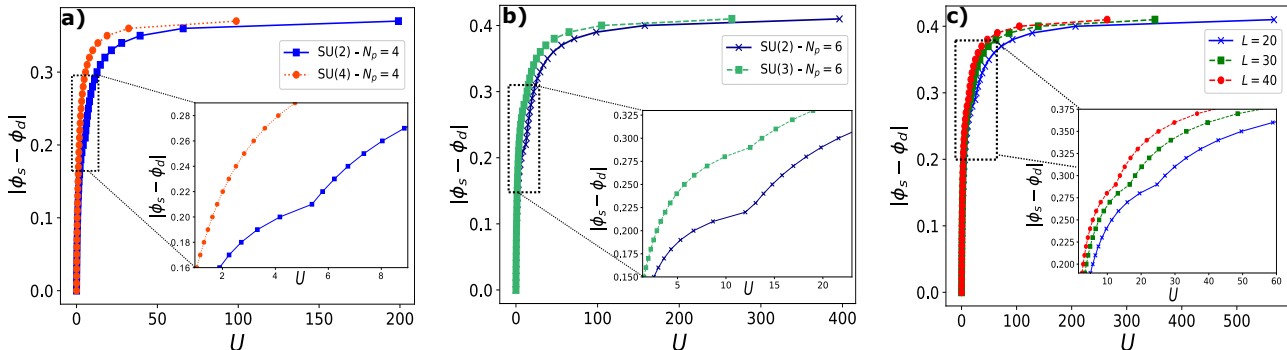

FIG. 9. Comparison of spinon creation in SU(2) fermions and SU($N$) fermions. a) Spinon creation flux distance $|\phi_s - \phi_d|$ against interaction $U$ is considered for a ring of $L = 40$ sites with $N_p = 4$ fermions with $N = 2$ and $N = 4$ spin components, where $\phi_s$ is the flux at which spinons are created and $\phi_d$ is the degeneracy point. The intermediate $U$ regime (inset) highlights the discontinuity present in the SU(2) case. b) Spinon creation flux distance against interaction for a ring of $L = 40$ sites with $N_p = 6$ fermions with $N = 2$ and $N = 3$ spin components. The inset depicts the disconituities for intermediate $U$ in both systems. **c)** Spinon creation is for a system with $N_p = 6$ particles with $N = 3$ with various system sizes, $L = 20$, $L = 30$ and $L = 40$. All the presented results are obtained with Bethe ansatz of the Gaudin-Yang-Sutherland model.

## PARITY EFFECT

Specific parity effects are observed for SU($N$) fermions. Both for commensurate and incommensurate fillings, the persistent current is found diamagnetic (paramagnetic) for ring systems containing $N_p = (2n+1)N$ ($N_p = (2n)N$) fermions, with $n$ being an integer. The nature of the current can be deduced by looking at the ground state energy of the system, whereby if the system has a minimum (maximum) at zero flux, then it is diamagnetic (paramagnetic) - Fig. 10. Such phenomena generalize the $4n/4n + 2$ of spin-$\frac{1}{2}$ fermions [57]. *Indeed, for both non-integer and integer fillings fractions, we demonstrate how results of Byers-Yang, Onsager and Leggett on the landscape of the system persistent current can be generalized to SU($N$) fermions* [37–39].

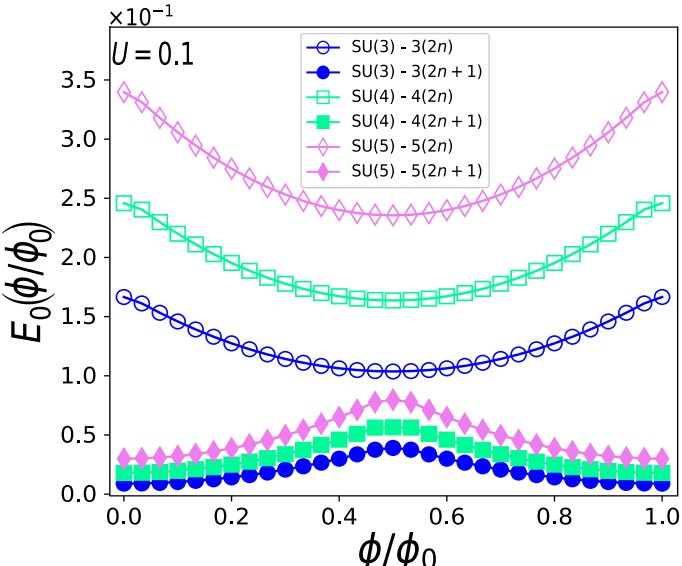

FIG. 10. Parity effect for SU($N$) fermions. Ground state energy $E_0(\phi)$ is plotted against the flux $\phi$ for different $N$ ranging from 3 (circles) to 5 (diamonds). Since the energy is suppressed (increased) by the effective magnetic field, systems with even (odd) number of particle per spin component are paramagnetic (diamagnetic). All the presented results are obtained by Bethe ansatz of Gaudin-Yang-Sutherland model for $L = 30$, with $N_p$ taken to be 1 particle and 2 particles per species for each $N$ corresponding to $n = 0, 1$ respectively.

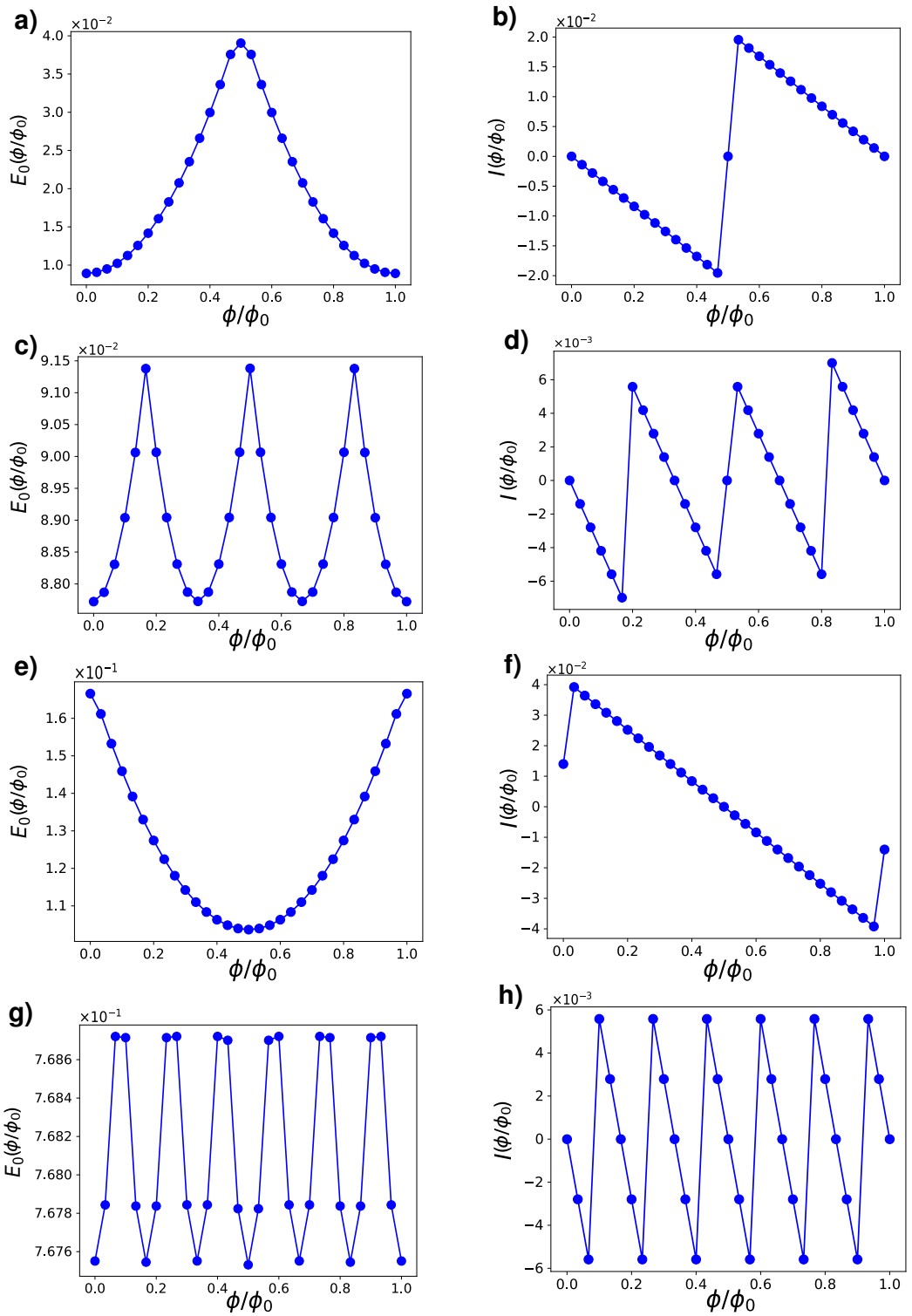

FIG. 11. SU($N$) persistent current and the corresponding ground state energy at incommensurate filling for different interaction strengths $U$. **a),b)** Ground state energy and Persistent current for $N_p = 3$ for $U = 0.1$. **c),d)** Ground state energy and persistent current for $N_p = 3$ for $U = 10,000$. **e),f)** Ground state energy and Persistent current for $N_p = 6$ for $U = 0.1$. **g),h)** Ground state energy and persistent current for $N_p = 6$ for $U = 10,000$. All curves are calculated with Bethe ansatz for the Gaudin-Yang-Sutherland model with $L = 20$.

The behaviour of this parity effect holds for small and intermediate $U$ but it is washed out at infinite $U$ for incommensurate fillings or above a finite threshold of interaction for integer fillings. Indeed, the character of the current is diamagnetic, since

the fractionalization of the bare flux quantum causes the ground state energy to always be a minimum at zero flux. In the cases where the current already had a diamagnetic nature at small values of $U$, its nature remains unchanged. The washing out of the persistent current can be clearly observed from Fig. 11 whereby comparison of SU(3) systems with $N_p = 3$ and $N_p = 6$ clearly show the stark difference in the nature of the current for the latter case between the different regimes of $U$.