# Peer review of "Persistent Current of SU(N) Fermions"

_SciPost Physics_

## Round 2 · Referee Report · Anonymous (Referee 2) · 2021-11-2

Strengths

1) the mechanism describing the emergence of spinon excitations in the ground state is new (to my knowledge) and interesting

2) the properties of the model are found by applying well-established theoretical and numerical techniques

Weaknesses

The definition of the term "mesoscopic" can be improved

Report

The authors have introduced several changes and new comments which increase the readability of this paper and improve the presentation of their work. In particular, the simple but useful comment concerning the role of BA quantun numbers in the emended version allows the reader to have an idea about the meaning of such important parameters. Also, the meaning for these authors of the term "mesoscopic" has been clarified. Reference [18] now allows the reader to get this information which the authors kindly gave to the referee in their response. Despite the presence of reference [18], however, I think that this simple and concise information (the fact that this paper considers mesoscopic effects namely effects that arise on length scales that are comparable with the particles’ coherence length) should be inserted in the paper 1) because there is no reason for omitting a simple but useful information, 2) because, as the authors probably know, the definition of "mesoscopic" is not univocal, 3) to better define the context of the entire work and 4) to avoid possible misunderstanding related to the use of different sizes of the system in the paper.

With this simple change I think that the emended version meets the acceptance criteria for publication on SciPost Physics.

Requested changes

Introduce the comment about mesoscopic effects that arise on length scales that are comparable with the particles’ coherence length

---

## Round 2 · Referee Report · Anonymous (Referee 1) · 2021-11-16

Report

The authors have satisfactorily address the issued that I raised in my previous report and therefore I recommend publication.

---

## Round 2 · Author Response

Dear Editor, many thanks for handling our manuscript “Persistent Current of SU(N) Fermions”. Many thanks also to the Referees for the their very good job in assessing our work.

Referee 1 finds our work a ‘very detailed study’ with ‘novel and interesting’ results and ‘relevant for future experi- ments’. The overall comment on our paper is ‘sufficiently high technical quality and the results are interesting enough to be granted publication’ after addressing the suggested revisions and comments.

Referee 2 finds the mechanism we discuss ‘new, nice and interesting’ and remarks on the solidity of our findings. On the impact, the Referee says: ‘I expect that the analysis developed in this paper should stimulate further theoretical work’. His recommendation is that: ‘This paper must be revised to improve its readability’

The Referee comments/criticisms prompted us improving on our work. In the following, we provide a point-to-point reply to the Referees’ comments and the list of changes made in the re-submitted manuscript.

Sincerely Yours, The Authors

REFEREE 1 1. R: “Page 2(LC). The long part from ”For N = 2 the BA ...” to ”... for finite value of U/t [4,26]” apparently contains a considerable amount of information. It supplies a list of properties and results relevant to model (1). This list, however, is almost disjoint from the subsequent discussion in the paper and its contribution to better understand the paper is almost null. The worst point concerns the comment ”The BA eigenstates are customarily labeled by a certain set of quantum numbers {Iα,Jβ}” and the subsequent comments. After noting that Iα and Jβj are undefined, the comment on 1) the zero-flux case and consecutive quantum numbers and 2) on the non-vanishing φ case where {Iα,Jβ} can change, turn out to be incomprehensible. Unfortunately, one discovers that these quantum numbers are repeatedly used in various important formulas in the sequel (for example for defining X and in eq. (2)). The authors must reorganize this part clarifying how the many properties/results listed are related to the following part of the paper. In particular, a clear definition (not relegated to the supplementary material) of the quantum numbers that characterize su(N) must be given.”

A: We note that the properties of the model we discuss in the Methods section are very important in the logic of our work: our approach is based on monitoring the exact diagonalization through Bethe ansatz. This granted us the access the quantitative analysis and the precise physical meaning of the various features of persistent current. In particular, through the quantum numbers labelling the eigenstates of the system, we are able to understand what is happening ’behind the scenes’ of the exact diagonalization. Nonetheless, we recognized that we could improve the presentation with the goal to make the manuscript self contained. Only for the very technical aspects of the Bethe ansatz, that would distract the reader from the physics, we referred to the appendix. Finally, for the interested readers, we provided general references on Bethe Ansatz analysis. Changes to manuscript 1.

  1. R: “The authors discuss the approach to model (1) in the integrable and non-integrable regimes. In the first case, in addition to the GYS-model regime, they consider the half-filling regime with large interaction captured by the Sutherland model (see page 2, (RC)). The latter, apparently, has nothing to do with the Lai-Sutherland model with nj = 1. This point should be clarified.”

A: We apologize for the oversight on our part and thank the Referee for pointing it out. The Referee correctly identifies that the two models do not refer to the same thing. We have addressed this by changing it to the Lai-Sutherland. Changes to manuscript 2.

  1. R: “The title of the section entitled ”Methods” seems quite inappropriate: only a few lines in the section are devoted to methods and techniques.”

A: As we mentioned above, in the Methods section we sketch the logic we employ for our analysis. The essentials of Bethe Ansatz are provided for the sake of characterizing the spectrum on physical grounds. Here, we note that, in our approach, the persistent current is a diagnostic tool that we use to probe the nature of the ground-state. 1

  1. R: “The term ”mesoscopic” is repeatedly used and seems to be an important aspect in this paper (but no comment is given in the text on this term). To have an idea of what it concretely means one must evince this information from the figure captions. The use of 30, 40 sites to get the results in figs. 1 and 2 is clearly related to the reduction of model (1) to the GYS model and than to the application of the BA analysis. But 30 or 40 sites represents (in many papers on lattice systems) macroscopic systems. Since L= 6, ... 18 is used to get the numerical results in figs. 3 and 4, the meaning of ”mesoscopic” is confused. This term must be carefully discussed in text and clarified. As an alternative, I do not exclude that the best thing to do is to remove ”mesoscopic” from the paper. In any case, a sufficiently extended comment on the values of L used in this paper for different regimes is necessary.”

A: Mesoscopic effects arise on length scales that are comparable with the particles’ coherence length (see for istance the book of Imry, our [18]). The persistent current is a standard probe for coherence in mesoscopic systems. It can be demonstrated on general settings that the persistent current amplitude scales as 1/L. This should be considered as the definition of what we mean by mesoscopic Changes to manuscript 3.

  1. R: “The study of a system in regimes with different filling should be made without changing the other model parameters. The fact that the system properties are derived by considering rather different sizes (see comment 4) of the lattice does not increase the scientific rigor of this paper. This is a delicate point that these authors have ignored but deserves some careful comment.” A: We thank the referee for this comment. The size of the system (L in our 1d system) is used as a knob to set the physics of the system in the BA accessible regimes (low filling or integer fillings and large interaction). At the same time, we change Np in order to point out how our phenomenon depends just on Np.

By considering the same number of particles but different system sizes, instead, one is able to clearly observe how the repulsive interactions between the particles affect the periodicity of the current.

Finally, we would like to note that we utilized Bethe ansatz results to corroborate those obtained by numerical means. As such, we considered system sizes where both the analytical and numerical results coincided with a high degree of accuracy.

REFEREE 2

  1. R: “On page 2, second column, near the bottom of the page, the authors state that they only consider systems with spin-singlet states where the total magnetization Sz = 0. They should either provide the definition of Sz operator for N > 2 or clarify this statement. I believe I understand what they mean by zero total magnetization, that is, the numbers of different species are all equal, but it is rather confusing in this context.”

For instance, for the sequence N = 2F +1 where F is the hyperfine spin of the alkaline-earth atom, Sz = Fz, the nuclear spin projection. However, this is not the case for values of N not being to this sequence as it is the case of N = 3, for which there are several possibilities of trapping three components exists.

A: We agree with the Referee that ’zero total magnetization’ is not the rigorous characterization of the states we consider in our work. Such jargon is imported from the N=2 case. For SU(N) one would need to consider N − 1 Cartan elements of SU(N). For example, in the case of SU(3) fermions, one would need to consider two Cartan elements that correspond (nA − nB ) and (nA + nB − 2nC ) where A, B, C denote the colours of the fermions. In order to avoid confusion and make the presentation clearer for the reader, we have addressed it by writing ‘Here, only systems with an equal number of particles per species are considered.‘ Changes to manuscript 4.

  1. R: “To be frank, I find Fig. 2 excessively crowded and not very clear. I think the authors should consider simplifying it, and if necessary split it into two separate figures. There is too much information on this figure that its main message is hard to grasp.”

A: We accept the comment of the Referee. We have substantially simplified the figure and made it more clearer. The caption was revised accordingly. Changes to manuscript 5. 2

  1. R: “On page 4, in the conclusions section, “The reduction of the effective flux quantum indicates that a form of ’attraction from repulsion’ can occur in the system. . . ” along with the following sentences seem to me rather unclear and citation of Refs. 9, 11, 37, 38 does not clarify the mechanism. If the authors have a clear physical picture they should explain it, otherwise they should remove this sentence(s).”

A: We agree with the Referee that this statement may have been unclear. We provided further explanation on this important effect. In previous studies it was demonstrated how the fractionalization in the persistent current is due to bound states formation. In such the bound states arise naturally since the considered systems were done of attracting bosons or fermions. In contrast, we consider fermions with repulsive interaction. The bound state probed by the persistent current has a different physical origin then: It is due to the spin degrees of freedom (coupled with the particle coherence). Consequently, in our system it looks as if an attraction arises from repulsion. Such feature is consistent with Kivelson and Chakravarty [47]. Changes to manuscript 5.

  1. R: “Although the analysis of the spinon creation in the ground state with flux and its relevance to the persistent current is very detailed and carried out from many different angles, I find the discussion regarding the connection to experiments is insufficient. For instance, due to the additional difficulties to cool fermions (as compared to bosons), finite temperature effects are always important. This implies that, by means of over the barrier thermal activation, the persistent current could decay and it would be useful if the authors could give some discussion of such effects (ideally some estimates to guide the experimentalists and give them an idea of the relevant time scale would be most welcome as well).”

A: We acknowledge the Referee’s comment. We added few comments on the possible experimental analysis that can be carried out through the analysis of the time-of-flight images. In particular, we commented on the impact of finite temperature effects. Furthermore, we included a reference to an experimental paper that was released earlier this year, that discusses in detail the decay of the persistent current. Changes to manuscript 7.

  1. R: “ Some remarks about the references: Personally, I do not like the current trend of citing only review articles and ignoring original references. For instance, Ref. [13] is a review article about the SU(N) symmetry exhibited by alkaline-earth gases. I think that, IN ADDITION to review articles, it would advisable to cite the original references where the existence of the symmetry was first pointed out: • AV Gorshkov, M Hermele, V Gurarie, C Xu, PS Julienne, J Ye, Zoller P, Demler E, MD Lukin and AM Rey Nat. Phys. 6 289 (2010) • MAC, A.F. Ho and M. Ueda New Journal of Physics 11, 103033 (2009) The same remarks apply to Ref. 21 about DMRG and possibly others. Finally, let me also point out that the journal name for Ref. 38 is missing.”

A: We apologize for the oversight on our part and thank the Referee for pointing this out. We have added the references where the symmetry was first pointed out and added the one for DMRG. Changes to manuscript 8.

---

## Round 2 · List of Changes

All changes carried out in the manuscript are highlighted red.
1. In line 1 of paragraph 2 on page 2, we changed “In our approach, the exact solution plays a very important role in classifying the eigenstates of the system. According to the general theory of BA solvable models, the spectrum of the model is obtained through the solution of coupled transcendental equations, which are paramterized by a specific set of numbers called the quantum numbers [28]. Indeed, they label all the excitations. For the specific case of the integrable SU(N) Hubbard models, the many-body excitations are labeled by quantum numbers: Ia, a = 1...Np and Jβj , βj = 1...Mj for j = 1...N − 1 with Ia and Jβj with being integers or half- odd integers, and Mj refers to the number of particles with a given component [20,28,29] (see Supplementary material). It is well known that, at zero flux φ, the ground state is found to be characterized by Ia = I1,I1 + 1,I1 + 2,...I1 + Np and Jβ1 = Jβ1,Jβ1 + 1,Jβ1 + 2,...Jβ1 + M1. Instead, sequences of Ia = I1 − 1, ∨,I1 + 1,I1+2,...I1+Np andJβj =Jβj −1,∨,Jβj +1,Jβj +2,...Jβj +Mj with‘holes’correspondtoexcitations; in particular holes in {Jβ} characterize the so-called spinon excitations [30]”

2. In line 4 of paragraph 2 on page 3, we changed “ half-filling & large interactions (captured by the Sutherland model)” to “a filling of one-particle per site & large interactions (captured by the Lai-Sutherland model)”.

3. We have added references 18, 19 and 32 to support our definition of mesoscopic. In line 4 of paragraph 3 on page 1 we added “which provides a standard avenue to probe the coherence of the system [19]”. Furthermore, in lines 5 and 9 of paragraph 3 on page 2, we added “Our diagnostic tool is the persistent current, providing access to the particles’ phase coherence [19,32]” and “The persistent current is of mesoscopic nature in that it corresponds to 1/L corrections of the ground state energy [19].”

4. In line 14 of paragraph 2 on page 3, we changed “Here, only systems with singlet states for which the total magnetization Sz = 0 are considered.” to “Here, only systems with an equal number of particles per species are considered.”

5. We simplified figure 2 and revised the caption accordingly, which now reads “a) Spinon creation flux distance |φs − φd| against interaction U. The inset contains the data in the intermediate U regime. b)Spinon creation flux distance against the interaction, rescaled by N and Np respectively, in the limit of low UNp. In this limit, spinon production is found to be a universal function of Np/N c) Spinon creation flux distance |φs −φd| against interaction per particle U/Np in the low U/Np regime. One can observe the enhancement of spinon production with increasing N.”

Furthermore we changed the text in line 4 of paragraph 2 on page 4 so as to make the figure easier to understand. We changed it to “Such values provide the number of spinons that can be present in the ground state at a given U. At moderate U, spinon production is found to be a universal function of the Np/N– see Fig. 2b; for systems with lower Np, spinons are generated at a lower value of interaction. For large U, spinon production is dictated by Np, with a fine structure that is determined by N: Systems with higher Np produce spinons at a lower value of U; for fixed Np, systems with the lower value of Np/N generate spinons at a lower U– see inset of Fig 2a. Such a phenomenon depends on the specific degeneracies of the system discussed previously, that facilitate spinon creation by increasing N. This feature emerges also by analysing the dependence of the phenomenon on the interaction per particle U/Np – Fig. 2c. We observe that N enhances the spinon production . While the number of spinons decreases with Np for N = 2, such a trend appears to be reversed for N > 2. For intermediate values of U, discontinuities arise in the curves in cases where Np/N > 1 (see Supplementary material). These discontinuities correspond to jumps ∆X in the spinon character X. By comparing systems with the same Np but different N, we note that the discontinuities tend to be smoothed out by increasing N and L (see Supplementary material). The value of ∆X results to be parity dependent.”

Lastly, we added a new figure (Figure 8) to the Appendix along with this text “A good way to visualize the impact of SU(N) fermions on spinon creation in the ground state is by having a figure of merit. By considering all the values of the flux φ at which spinons can be created and noting down the value of the interaction U, at which there is a level crossing between the ground state and an excited state, one can obtain this figure of merit. The curves displayed in Figure 8 are obtained by monitoring the spinon flux distance |φs − φd| at which there are crossings between states with spinons and those without, with φs being the flux at which spinons are created and φd being the degeneracy point. Figure depicts the spinon creation flux distance against the interaction per particle U/Np. The inset denotes the data in the intermediate U/Np regime. Due to the specific N − 1 types of excitations that are inherently present in SU(N) fermions for N > 2, spinon creation is facilitated with N. ”.

6. In line 5 of paragraph 1 on page 3, we added “In particular, the periodicity of the persistent current reflects the structure of the ground-state. For example in the case of a BCS ground-state, the period of the persistent current is halved due to the formation of Cooper pairs [37,40]. Similarly for bosonic systems, the persistent current fractionalized with a period of 1/Np, indicating the formation of a bound state of Np particles [41]. The specific parity effects also hold for SU(N) systems and SU(2) fermions [37,38].”

7. In line 5 of paragraph 2 on page 5, we added “Recently, the persistent current of SU(2) fermions was experimen- tally realized [52].” In line 11 of paragraph 2 on page 5, we added “Finite temperature is expected to reduce the visibility of the time of flight images. Nonetheless, based on the predictions for N = 2, the main features of the persistent current are expected to be feasibly detectable [53]. The temperature can activate transitions between different angular momentum states, thereby causing a decay in the amplitude of the persistent current [54]. Recently, the dependence of persistent currents on finite temperature effects has been investigated for SU(2) fermions [55]. However, the actual decay of the persistent current of SU(N) fermions needs an in-depth study.”

8. We have added the recommended references by the referee, which are references [3] and [4]. We have also put in the original DMRG reference, which is reference [25]. Lastly, we put in the Journal name for reference [49].

---

## Editorial Decision

resubmitted